# A study of systemic risk spillovers in Asian emerging markets and Chinese stock market

Zhongzheng Fang [1, 2*]

1 Faculty of Business Administration, Kangnam University, Yongin-si, Gyeonggi-do, Republic of Korea,
2 Division of Global Studies, Sungkonghoe University, Seoul, Republic of Korea

* fzz123760@naver.com

## Abstract

This study examines systemic risk spillover effects between China's Shanghai Stock Exchange (SSE) and seven Asian emerging markets within the context of increasing global financial integration. Utilizing Quantile Regression and Conditional Value-at-Risk (CoVaR) methodologies, this study provides a new perspective on understanding the asymmetry of systemic risk transmission between China and Asian emerging markets. Based on data from 2000 to 2024, the findings reveal significant spillover patterns, with Korea (KOSPI) showing high sensitivity to SSE risks, Malaysia (KLCI) exerting strong influence, and Thailand (SET) and Taiwan (TWII) emerging as key contributors and receivers of systemic risk. Under extreme market conditions, risk spillovers intensify, positioning SSE as a central hub in regional risk dynamics. These insights underscore the need for robust macroprudential policies and enhanced regional cooperation to mitigate systemic vulnerabilities, contributing to both the theoretical discourse on financial risk and its practical management.

## 1. Introduction

As global financial markets continue to integrate, China's financial markets, which have been progressively opening up, have evolved into vital players in Asia and worldwide. Year by year, China's capital market has expanded, along with the concurrent enrichment of financial products and services, thereby offering increased investment opportunities for domestic and foreign investors. In this process, the stability and prosperity of China's financial markets have positively influenced Asia and global financial markets [1].

Against this backdrop, it is particularly important to explore the systemic risk spillovers between the fast-growing Asia emerging markets, which are now integrated with the global financial system and the China stock market. The transmission of financial market shocks between Asia emerging markets and China stock markets has become an important area of research. Most existing studies focus on the financial contagion among developed markets [2], and relatively little research has been conducted on the risk contagion among emerging markets [3].

**Data availability statement:** The raw data supporting the conclusions of this article have been made available by the authors, without undue reservation, using the Harvard Dataverse repository. (via: https://doi.org/10.7910/DVN/3QY4JA, Harvard Dataverse, V2).

**Funding:** The author(s) received no specific funding for this work.

**Competing interests:** The authors have declared that no competing interests exist.

Currently, China, as the second largest economy, is increasingly linked to emerging Asia markets, as these markets are closely related through trade and financial flows [4]. Therefore, volatility in China's stock market could affect other markets through a variety of channels, such as direct trade links, capital flows, and investor confidence. Likewise, China markets can be affected by other Asia emerging markets. This impact may be reflected not only in the linkage of market yields but also in the linkage of risks with other needs, i.e., systemic risk spillovers.

The Asian emerging markets selected for this study are based on MSCI's annual global market classification for 2024 (https://www.msci.com/our-solutions/indexes/market-classification) and cover seven representative markets: India (BSE), Indonesia (JKSE), Taiwan (TWII), Malaysia (KLCI), the Philippines (PSI), Thailand (SET) and South Korea (KOSPI). These markets are defined by MSCI as Emerging Markets, characterized by high economic growth potential, developing capital markets, and increasing dependence on the global financial system. These markets have particularly strong geo-economic and financial ties to China, making them ideal for studying systemic risk spillovers between China and emerging markets in Asia. MSCI's market classification criteria not only consider economic scale and growth potential but also emphasize market liquidity, foreign investment accessibility, and the robustness of capital market infrastructure [5,6]. Due to capital controls and insufficient market liquidity, countries such as Pakistan and Bangladesh are excluded from this study. Similarly, Vietnam is classified as a Frontier Market by MSCI, primarily due to its restrictions on foreign capital access and lower market liberalization, making it less suitable for assessing systemic risk spillovers in emerging markets.

Furthermore, this study focuses on systemic risk transmission between China and Asian emerging markets, which necessitates selecting markets with strong trade, investment, and capital flow linkages to China. The MSCI-classified Asian emerging markets are particularly sensitive to China's financial and economic fluctuations, making them more appropriate for analyzing China's risk spillover effects in the region.

In recent years, Asia emerging markets have experienced rapid economic growth and financial market development, but at the same time, they have exposed the problem of systemic risk. For example, the Asia financial crisis, global financial crisis, COVID-19 pandemic, Russia–Ukraine war, and other incidents have seriously impacted Asia emerging markets. Therefore, studying the interaction mechanism between the China stock market and the systemic risks in emerging Asia markets is of great theoretical and practical significance for an understanding of the risk propagation mechanism in financial markets, for warnings related to financial risks, and for the formulation of effective macroprudential policies.

This study provides an in-depth analysis of the systematic risk spillovers between the China stock market and seven emerging Asian markets using quantile regression modelling and conditional value-at-risk (CoVaR) methodology. The findings of this study have important practical implications for investors, policymakers and

financial regulators. The paper is structured as follows: Section 2 provides a review of the relevant literature. Section 3 describes in detail the methodology used in the study. Section 4 presents our data and the empirical results derived from the analysis. Finally, Section 5 concludes the paper with a summary of the main insights and points of the study.

## 2. Literature review

### 2.1. Theories of spillover effects

The spillover effect between financial markets is based on two types of spillover effects: the spillover effect of the mean and the spillover effect of the variance. With the deepening of risk management practice, more attention has been focused on the spillover effect of the variance (i.e., volatility spillover effect). Shen et al. [7] defined the risk spillover effect concept on the basis of the risk transmission mechanism among different capital markets. Furthermore, Yadav et al. [8] argued that the yield volatility caused by the difference in asset prices among different capital markets is transmitted from one country's capital market to that of other countries' capital markets. Other scholars have suggested that risk spillover effects occur when shock and volatility in the capital market of one country have an impact on the capital markets of other countries with which its economy is closely linked [9,10].

### 2.2. Methodologies for risk transmission analysis

Bekaert et al. [11] examined the impact of the 2008 global financial crisis on global equity markets. Although the crisis originated in the United States, it was catastrophic for stock markets, including both emerging and developed markets. By applying Granger [12] causality tests and risk measures based on GARCH models, the authors revealed the propagation mechanism of the financial crisis. Their findings suggested that financial crises triggered risk spillovers in the stock market, i.e., the risky state of one market affected the other markets. The risk correlation among individual markets may increase during a financial crisis or other significant economic events, thus triggering systemic risk spillovers. Other scholars performed relevant studies on the basis of these econometric models and the Granger causality model [13], as well as within the vector autoregressive (VAR) framework [14–16]. However, these approaches often assume linear dependencies, limiting their ability to capture tail risks and asymmetric relationships in emerging markets.

In financial risk management, it is critical to understand and monitor the health of financial institutions. In particular, when a financial institution is in a poor condition, it can significantly impact the entire financial system, triggering systemic risk. Therefore, Adrian and Brunnermeier [17] introduced the concept of *CoVaR* (Conditional Value-at-Risk), a new risk measure for financial systems. Laeven et al. [18], using $\triangle CoVaR$ and other methods, determined the relationship among bank size, capital, and systemic risk, which showed that the larger the asset size of a bank, the greater the degree of its contribution to systemic financial risk.

Wang et al. [19] investigated the tail risk spillover effects among international financial markets using quantile vector autoregressive (QVAR) models. Their findings showed that the risk spillover effects among individual financial markets were significantly enhanced under extreme market conditions. The main content of the paper explored the quantile-based spillovers in 17 stock markets from January 1993 to January 2022. In contrast to the traditional mean-based spillover measure, this new quantile approach allowed for a detailed investigation of the spillover in each quantile and captured the spillovers under extreme event conditions.

Based on these insights, Meng and Chen [20] emphasize that a certain degree of correlation exists between individual markets within the financial system. This correlation facilitates the transmission of shocks, leading to volatility spillovers rippling through interconnected markets. Together, these studies emphasize the importance of adopting advanced methodologies (e.g., quantile-based approaches) to better understand the dynamics of systemic risk propagation, especially in tail risk scenarios.

The literature on financial market spillovers has evolved significantly, integrating theoretical and empirical advancements. Beyond mean and volatility spillovers, recent studies emphasize alternative risk measures for systemic risk transmission. Diebold and Yilmaz [15] developed a network topology model using directed acyclic graphs to capture financial market interconnectedness, identifying key systemic nodes in risk propagation. Similarly, Baruník and Křehlík [21] introduced a frequency dynamics spillover approach, distinguishing between short- and long-term spillover effects, providing deeper insights into market shock transmission mechanisms. These advanced econometric models highlight the need for non-traditional approaches in financial risk assessment.

## 2.3. Empirical findings on Asian markets

An increasing body of research explores the role of macroeconomic factors in shaping financial market spillovers. Kannadhasan and Das [22] analyzed and compared the impact of economic policy uncertainty (EPU) and geopolitical risk (GPR)-related shocks on emerging Asian equity markets. Their findings suggest that increased uncertainty amplifies spillover effects, particularly in emerging markets, which are more exposed to external shocks. Monetary policy decisions play a crucial role. Ehrmann and Fratzscher [23] showed that central bank interest rate changes, especially by the Federal Reserve, trigger global financial market reactions, affecting capital flows and asset price volatility. Gagnon et al. [24] further demonstrated that unconventional monetary policies, such as quantitative easing, heighten financial integration and increase spillover intensity across international markets.

Recent studies highlight the unique vulnerabilities and transmission mechanisms of financial spillovers in emerging markets. These markets tend to exhibit heightened sensitivity to global financial disruptions due to capital flow volatility, regulatory constraints, and limited market depth. For example, Sula [25] found that emerging economies experience stronger contagion effects during global financial instability, particularly when capital flight and exchange rate pressures increase. Similarly, Azad and Serletis [26] demonstrated that U.S. monetary policy spillovers disproportionately impact emerging markets under conditions of heightened uncertainty.

Khawaja [27] emphasized emerging markets' growing importance in global financial networks due to their rapid economic growth, expanding capital markets, and increasing foreign investment. However, their structural fragility, as discussed by Karanasos et al. [28], amplifies systemic risk compared to developed economies. Akyüz [29] found that the structure of commodity trade in commodity-dependent developing countries plays a significant role in shaping their financial vulnerability. Additionally, Wang and Wang [30] analyzed the interactions between BRICS stock markets and cryptocurrencies, finding increased financial interdependencies during market turbulence. Moreover, most Asian equity and foreign exchange markets are net risk receivers [31]. Feng et al. [32] study the systematic risk spillovers from the Belt and Road equity markets between 2008 and 2021, and analyse that the network effects are asymmetric and regionally heterogeneous across markets at different risk levels.

## 2.4. Limitations of previous studies

Despite the recognition of systemic risk and financial contagion in emerging markets, most existing studies focus on developed markets or rely on traditional econometric techniques that assume linear dependence. While some recent studies (e.g., Wang et al. [19]) have explored quantile-based spillovers, research specific to Asian emerging markets remains limited, and the dynamics of risk spillovers under extreme market conditions require further exploration. Given Asia's increasing financial integration and evolving market structure, a more refined understanding of risk spillover dynamics is essential.

Moreover, existing spillover models often fail to fully capture risk transmission under extreme market conditions. Traditional models such as VAR and GARCH can effectively model mean spillovers but lack the granularity needed to assess tail risk and asymmetric dependence. Most prior research typically treats systemic risk spillovers as mutually reinforcing relationships, without analyzing whether one market acts as a dominant risk transmitter while others function as net receivers.

To address these gaps, this study employs quantile regression and CoVaR methods to distinguish between China's impact on Asian emerging markets and the influence of Asian markets on China. This approach provides a detailed understanding of asymmetric systemic risk propagation, highlighting the directionality and magnitude of risk transmission between markets. By focusing on these interactions, this study aims to provide new insights into Asia's role in global financial stability and to inform policymakers in developing tailored risk management strategies in an increasingly interconnected world.

## 3. Methodology

### 3.1. CoVaR methodology

"Value-at-Risk" (VaR) was introduced by JP Morgan Chase in the 1990s, and it revolutionized risk measurement theory and practice. Consequently, it has become a mainstream technique for risk management and has been widely adopted by major financial institutions and financial regulators. As risk management practice advanced, it became clear that VaR had its own limitations [33]. Most importantly, it could speculate on the potential risk of a portfolio of marketable securities only under "normal" market conditions [34] but not under extreme market conditions. The most important characteristic of a financial crisis is the rapid spread of losses throughout financial institutions (or financial markets), which ultimately leads to systemic risk and disruption in the financial system.

The *CoVaR* approach has a very broad applicability, as it integrates risk transfer effects into the VaR approach. In *CoVaR*, Co refers to "conditional", "peer (simultaneous)", "contagious", etc.; thus, *CoVaR* can be understood as the "conditional VaR" experienced by other financial firms or the system as a whole when a particular financial institution is in crisis. *CoVaR* also implies the risk contribution of individual financial firms to the financial system as a whole or to a particular financial institution. In this study, we developed the *CoVaR* and $\Delta$ *CoVaR* equations using the research models of Adrian and Brunnermeier [17] and Tobias and Brunnermeier [35].

The VaR-based risk management measure is defined as the maximum loss that could occur in a financial institution (market) at a particular confidence interval $q$ (99%, 95%, or 90%). $X^i$ is the value level of a financial institution (market) $i$, and $VaR_q^i$ is the maximum loss that could occur in a financial market $i$ at the probability level of $1 - q$, expressed as a negative value:

$$\Pr\left(X^i \leqslant VaR_q^i\right) = q.$$

(1)

The traditional VaR model ignores the risk spillover effects among financial institutions (markets), which can greatly reduce the accuracy of the risk assessment. Hence, the *CoVaR* model, based on the VaR model, was established. Adrian and Brunnermeier [17] developed its modified definition as $CoVaR_q^i$, which represents the level of risk faced by a financial institution (market) $j$ when a financial institution (market) $i$ is in an extremely unfavorable value-at-risk state. Unlike traditional VaR, CoVaR combines dynamic interdependence and systematic importance, making it ideal for the asymmetric and nonlinear risk structures observed in emerging markets.

$$\Pr\left(X^j \leqslant CoVaR_q^{j|C(X^i)} \mid C\left(X^i\right)\right) = q,$$

(2)

where $C(X_i)$ denotes the current state of extreme risk X of a financial institution (financial market) $i$, which means that $X^i$ is in the extreme case of $VaR_q^j$. In order to better reflect how the risk of a financial market $j$ changes when a financial institution (market) $i$ moves from the normal state to the risky state, i.e., the size of the risk spillover, the static $\Delta CoVaR_q^{j|i}$ of risk spillover effect is defined as follows:

$$\Delta CoVaR_q^{j|i} = CoVaR_q^{j|i} - VaR_q^i$$

(3)

To further reflect the degree of the risk spillover from $i$ to $j$, it is necessary to normalize this expression, thus obtaining

$$\%CoVaR_q^{j|i} = \left( \Delta CoVaR_q^{j|i}/VaR_q^j \right) \times 100\ \%$$

(4)

### 3.2. Quantile regression measures CoVaR values

Traditional regression analysis is used primarily to study the relationship between the dependent variable and the independent variables. However, traditional least squares regression can be based only on the conditional mean and the conditional median at the center of the distribution, which can prevent researchers from providing a complete description of the conditional distribution. [36]. Quantile regression was first proposed by Koenker and Bassett [37] in 1978. This method studies the relationship among the conditional quartiles of the independent and dependent variables; its utility lies in the fact that it does not require specific distributional assumptions, is not restricted to a specific model, can solve the extreme value problem well, and ensures the robustness and validity of the regression model. Quantile regression is a method that estimates the linear relationship between one independent variable and one dependent variable quantile. The quantile regression method may be more effective than the least squares method [38]. According to the definition of the *CoVaR* model, this study used the quantile regression method to measure the *CoVaR* value. The following $q$-quantile regression model was first developed:

$$\hat{R}_q^{j|i} = \hat{\alpha}^i + \hat{\beta}^i R^i$$

(5)

where $R_q^{j|i}$ denotes the quantile value of a financial institution (market) $j$ at confidence level $q$ when risk time occurs in a financial (institutional) market $i$. According to the VaR definition, we can obtain the following:

$$VaR_q^j \mid R^i = \hat{R}_q^{\frac{j}{i}}$$

(6)

Under the quantile regression model, *CoVaR* values can be estimated using the coefficient estimates $\hat{\alpha}^i$ and $\hat{\beta}^i$.

$$CoVaR_q^{\frac{j}{i}} = VaR_q^j \mid VaR_q^i = \hat{\alpha}^i + \hat{\beta}^i VaR_q^i$$

(7)

### 3.3. Quantile regression and the dynamic *CoVaR* model

The contribution of individual financial institutions to the systemic risk calculated by the static *CoVaR* model does not change over time; it is only an overall description. To investigate the dynamics of the systemic risk contribution of different financial institutions, this study introduces the *VaR(q)* quantile q = 0.05, which is the *VaR*(0.05) of an institution $i$ under normal conditions. The $\Delta CoVaR$ is expressed as follows:

$$\Delta CoVaR_{q,t}^{j|i} = CoVaR_{q,t}^i - CoVaR_{0.05,t}^i = \hat{\beta}_q^{j|i} \left( VaR_{q,t}^i - VaR_{0.05,t}^i \right)$$

(8)

The CoVaR methodology, as a conditional value-at-risk (VaR) measure, has been used to assess the maximum loss that could be suffered by other financial institutions or the financial system as a whole if a particular financial institution or market is in a state of crisis. This approach captures risk propagation and interdependence within the financial system and fills the gap where the traditional VaR approach ignores systemic risk transmission effects [39]. Application and advantages of quantile regression provide an analytical tool that goes beyond traditional regression models by allowing the researcher to

assess the relationship between different quartiles of the distribution of returns on financial assets, not just the conditional mean. This approach is particularly well suited to analysing extreme events and tail risks in financial markets, as it does not rely on data-specific distributional assumptions, increasing the robustness and flexibility of the model [40].

By combining CoVaR and quantile regression methods, this study is able to identify and measure the propagation mechanisms of systemic risk more precisely, especially under extreme market conditions. This choice of methodology is superior to other approaches as it provides a deeper and more comprehensive perspective on the dynamics of risk propagation in financial markets and contributes to more effective management and mitigation of systemic risk. The author declares that this study complies with all applicable ethical research standards.

## 4. Empirical analysis

For this study, India (BSE), Indonesia (JKSE), China's Shanghai Stock Exchange (SSE), Taiwan (TWII), Malaysia (KLCI), the Philippines (PSI), Thailand (SET), and South Korea (KOSPI) were selected as the Asia emerging markets. The data source used was DataGuide5 (https://dataguide.fnguide.com), and the sample interval was selected from 2000.01.01 to 2024.5.31, with closing prices for five trading days per week. Because the indexing methods could differ across markets, the data were used to provide consistency and to meet the requirements of the econometric model for time series smoothness. The closing price data of each financial market were transformed into log returns and then magnified 100 times to enhance the readability of the data results. $R_t$ is the stock return on day $t$; $P_t$ and $P_{t-1}$ are the closing prices of stocks on day $t$ and $t-1$, respectively:

$$R_t = 100 \times Ln(\frac{P_t}{P_{t-1}})$$

(9)

Table 1 shows the results of the descriptive statistics for the eight financial markets. It can be seen that the skewness of the return series of the eight financial markets are all less than 0, i.e., they are left-skewed distributions. The kurtosis is greater than 3, and all of them are steeper than the normal distribution. In summary, all series show the characteristics of spikes and thick tails. The p-values of the Jarque–Bera test are all 0, indicating that the hypothesis of a normal distribution is rejected for each return series. Therefore, the data in this paper are suitable for study by quantile regression.

The ADF (Augmented Dickey–Fuller) method was used to conduct a unit root test for the eight time series in order to verify further the smoothness of each series. As shown in Table 2, each t-statistic is smaller than the level value at the significance levels of 1%, 5%, and 10%, and the p-value is 0. Therefore, the original hypothesis that there is a unit root in each return series is rejected, i.e., there is no unit root. It is, thus, proved that all time series are smooth and can be analyzed using quantile regression and the *CoVaR* model.

**Table 1. Sample descriptive statistics.**

|  | BSE | JKSE | KLCI | KOSPI | PSI | SET | SSE | TWII |
|---|---|---|---|---|---|---|---|---|
| Mean | 0.042 | 0.037 | 0.010 | 0.015 | 0.018 | 0.016 | 0.013 | 0.014 |
| Maximum | 15.990 | 9.704 | 6.626 | 11.284 | 16.178 | 9.639 | 9.034 | 6.285 |
| Minimum | -14.102 | -9.803 | -6.811 | -12.805 | -14.322 | -12.592 | -8.873 | -8.302 |
| Std. Dev. | 1.288 | 1.114 | 0.723 | 1.419 | 1.168 | 1.264 | 1.319 | 1.210 |
| Skewness | -0.817 | -0.622 | -0.516 | -0.518 | 0.027 | -0.701 | -0.312 | -0.563 |
| Kurtosis | 17.161 | 11.358 | 12.030 | 10.488 | 23.021 | 12.827 | 10.134 | 8.453 |
| Jarque–Bera | 52623.590 | 18492.450 | 21389.530 | 14797.320 | 103805.400 | 25515.440 | 13278.46 | 8029.043 |
| Probability | 0.000 | 0.000 | 0.000 | 0.000 | 0.000 | 0.000 | 0.000 | 0.000 |
| Obs. | 6214 | 6214 | 6214 | 6214 | 6214 | 6214 | 6214 | 6214 |

The risk faced by the other financial markets (*i*) was calculated separately when the stock market in China's Shanghai Stock Exchange (SSE) was in distress, as was the risk faced by China's Shanghai market when each of the remaining seven markets was in distress. A quantile regression model at q = 0.05 was developed, as in Equations (10) and (11). To calculate the value of *CoVaR*, a quantile regression was first performed to estimate the parameters $\alpha$ and $\beta$. As shown in Table 3, the coefficient estimates obtained from the 5% quantile regression were calculated by python. $\beta_1$ represents the spillover effect of each financial market on the marginal systemic risk of the China Shanghai market, reflecting the sensitivity of the China Shanghai market return to the changes in the returns of each financial market. The $\beta_2$ coefficients represent the impact of the China's Shanghai Stock Exchange (SSE) on the systemic risk of other Asian emerging markets. Each estimate was positive, indicating that changes in the SSE's systemic risk directly align with risk changes in other markets. An increase in risk in the SSE or an increase in risk in these markets consistently heightened the systemic risk across the region. Comparing $\beta_1$ and $\beta_2$ reveals nuanced relationships between the SSE and other markets.

For India (BSE), the results show that $\beta_1$ (0.232) and $\beta_2$ (0.244) are nearly equal, suggesting that the systemic risk influence between the SSE and the BSE is symmetric. Both markets demonstrate a balanced reciprocal impact, indicating mutual susceptibility during risk events. In contrast, for the Korea (KOSPI) market, $\beta_1$ (0.214) is less than $\beta_2$ (0.317), indicating that the influence of the SSE on KOSPI's risk is greater than the reverse. This asymmetric relationship highlights the SSE's more significant role in systemic risk propagation for the Korean market. For the five other emerging markets—Indonesia (JKSE), Taiwan (TWII), Malaysia (KLCI), the Philippines (PSI), and Thailand (SET)—the relationship between $\beta_1$ and $\beta_2$ reveals a distinct pattern. In all five cases, $\beta_1 > \beta_2$, indicating that the degree of influence these markets exert on SSE's risk is greater than the reverse. For example, Malaysia (KLCI) demonstrates the highest $\beta_1$ (0.513) among all markets, reflecting its strong influence on SSE, whereas its $\beta_2$ (0.138) is the lowest, indicating limited systemic feedback from SSE to KLCI.

The sensitivity ranking of these markets to systemic risk in the SSE, based on $\beta_2$, is as follows: Korea (KOSPI)> Taiwan (TWII)> Indonesia (JKSE)> India (BSE)> Thailand (SET)> the Philippines (PSI)> Malaysia (KLCI). This order suggests

**Table 2. Unit root test.**

|  | BSE | JKSE | KLCI | KOSPI | PSI | SET | SSE | TWII |
|---|---|---|---|---|---|---|---|---|
| t-statistic | -72.834 | -66.715 | -48.417 | -78.361 | -69.167 | -53.456 | -36.427 | -51.899 |
| Prob. | 0.000 | 0.000 | 0.000 | 0.000 | 0.000 | 0.000 | 0.000 | 0.000 |
| Test critical values: |  |  | 1% level |  |  | -3.431 |  |  |
|  |  |  | 5% level |  |  | -2.862 |  |  |
|  |  |  | 10% level |  |  | -2.567 |  |  |

**Table 3. Quantile regression equation coefficients.**

|  | BSE | JKSE | KLCI | KOSPI | PSI | SET | TWII |
|---|---|---|---|---|---|---|---|
| $\alpha_1$ | -1.965 | -1.976 | -1.971 | -1.955 | -1.987 | -1.975 | -1.920 |
| t-statistic | -26.445 | -30.143 | -32.595 | -36.618 | -34.554 | -33.993 | -33.295 |
| $\beta_1$ | 0.232 | 0.333 | 0.513 | 0.214 | 0.259 | 0.370 | 0.349 |
| t-statistic | 2.773 | 3.890 | 4.339 | 4.113 | 3.250 | 5.894 | 5.211 |
| $\alpha_2$ | -1.919 | -1.722 | -1.092 | -2.174 | -1.711 | -1.899 | -1.875 |
| t-statistic | -42.696 | -32.022 | -40.763 | -36.952 | -36.487 | -45.316 | -37.834 |
| $\beta_2$ | 0.244 | 0.261 | 0.138 | 0.317 | 0.163 | 0.239 | 0.282 |
| t-statistic | 5.124 | 4.416 | 4.735 | 5.430 | 3.055 | 5.169 | 5.102 |

Note: ( ) represents the t-statistic.

that Korea and Taiwan are more vulnerable to systemic risk transmitted from the SSE compared to other markets. Conversely, the influence of these markets on the SSE, based on $\beta_1$, follows the order: Malaysia (KLCI)> Thailand (SET)> Taiwan (TWII)> Indonesia (JKSE)> India (BSE)> the Philippines (PSI)> Korea (KOSPI). This ranking highlights Malaysia and Thailand as significant contributors to systemic risk impacting the SSE. In summary, the analysis underscores the asymmetric nature of risk propagation across the region, with the SSE exerting a stronger influence on some markets (e.g., Korea, Taiwan) while being more influenced by others (e.g., Malaysia, Thailand).

$$R^{SSE}_{0.05} = \hat{\alpha}_1{}^{SSE|i} + \hat{\beta}_1{}^{SSE|i}R_{0.05}{}^{i} \tag{10}$$

$$R^{i}_{0.05} = \hat{\alpha}_2{}^{i|SSE} + \hat{\beta}_2{}^{i|SSE}R_{0.05}{}^{SSE} \tag{11}$$

In this analysis, this paper used the values of parameters $\alpha$ and $\beta$ obtained from the quantile regression to calculate the *CoVaR* values—those of each emerging market and the China Shanghai market for a given VaR of the China Shanghai market and of each emerging market, respectively. Which are calculated with reference to equations (12, 13, 14, 15, 16)

$$CoVaR^{i}_q = VaR^{i}_q \mid VaR^{SSE}_q = \hat{\alpha}^{SSE} + \hat{\beta}^{SSE}VaR^{SSE}_q \tag{12}$$

$$CoVaR^{SSE}_q = VaR^{SSE}_q \mid VaR^{i}_q = \hat{\alpha}^{i} + \hat{\beta}^{i}VaR^{i}_q \tag{13}$$

$$\Delta CoVaR^{i}_q = CoVaR^{i}_q - VaR^{SSE}_q \tag{14}$$

$$\Delta CoVaR^{SSE}_q = CoVaR^{SSE}_q - VaR^{i}_q \tag{15}$$

$$\%CoVaR^{SSE}_q = \left( \Delta CoVaR^{SSE}_q / VaR^{i}_q \right) \times 100\,\% \tag{16}$$

In Table 4, terms of individual risk (*VaR*), KOSPI (-2.270) and SSE (-2.035) exhibit the highest VaR, suggest that the South Korean and Chinese markets are more sensitive to extreme market conditions, possibly due to their reliance on global manufacturing supply chains and high exposure to external economic shocks [41]. BSE (-1.937), SET (-1.930) and TWII (-1.927) have moderate *VaR*, indicating their dual role as risk transmitters and receivers during systemic events. The intermediate levels of *VaR* for these markets likely reflect their integration with global trade while maintaining some degree of financial insulation [9]. In contrast, JKSE (−1.723) and PSI (−1.715) have relatively low VaR values, demonstrating the stability of the Indonesian and Philippine markets. This can be attributed to lower market volatility and lesser reliance on external capital [42]. KLCI (−1.088), with the lowest VaR, highlights Malaysia's remarkable market stability, supported by higher levels of capital controls and reduced exposure to foreign exchange fluctuations, which buffer against global systemic risks.

Systemic risk diffusion through $|CoVaR| > |VaR|$ shows the amplification effect of markets in systemic events, a phenomenon that is particularly pronounced in emerging markets. Among them, SET (-2.689) and TWII (-2.593) contribute the most to the systemic risk of SSE, this can be attributed to Thailand and Taiwan's extensive trade and supply chain linkages with the Chinese market, which exacerbate risk spillovers during systemic events [43], while KOSPI (-2.441) and BSE (-2.414) have relatively small impacts, suggesting that the South Korean and Indian

**Table 4. Comparative analysis of the contribution of systemic risk.**

|  | BSE | JKSE | KLCI | KOSPI | PSI | SET | TWII |
|---|---|---|---|---|---|---|---|
| $VaR^{SSE}_{0.05}$ | -2.035 | -2.035 | -2.035 | -2.035 | -2.035 | -2.035 | -2.035 |
| $VaR^{i}_{0.05}$ | -1.937 | -1.723 | -1.088 | -2.270 | -1.715 | -1.930 | -1.927 |
| $CoVaR^{SSE}$ | -2.414 | -2.550 | -2.530 | -2.441 | -2.432 | -2.689 | -2.593 |
| $CoVaR^{i}$ | -2.415 | -2.254 | -1.373 | -2.819 | -2.043 | -2.387 | -2.449 |
| $\Delta CoVaR^{SSE}$ | -0.379 | -0.515 | -0.494 | -0.406 | -0.397 | -0.655 | -0.558 |
| $\Delta CoVaR^{i}$ | -0.478 | -0.531 | -0.285 | -0.549 | -0.328 | -0.456 | -0.523 |
| $\%CoVaR^{SSE}$ | 18.624 | 25.307 | 24.303 | 19.957 | 19.505 | 32.174 | 27.421 |
| $\%CoVaR^{i}$ | 24.673 | 30.803 | 26.163 | 24.196 | 19.144 | 23.635 | 27.122 |

Note: ( ) denotes rank order of risk contribution.

markets are more likely to be connected to the Chinese market through indirect channels Associations. On the other hand, the impact of SSE on other markets (CoVaR^i) is strongest for KOSPI (-2.819), with TWII (-2.449) and BSE (-2.415) also significantly affected; while KLCI (-1.373) shows the lowest risk propagation, thanks to Malaysia's diversified economic structure [44].

ΔCoVaR and %CoVaR reveal the relationship of systemic risk propagation between the Chinese market and other emerging markets. In terms of ΔCoVaR, SET (-0.655) has the highest incremental exposure to SSE, indicating that its volatility has the strongest amplifying effect on the Chinese market. TWII (-0.558) and JKSE (-0.515) follow closely, reflecting the important impact of Taiwan and Indonesia markets. PSI (-0.397) and BSE (-0.379), contribute the least to the incremental exposure to the Chinese market, suggesting weaker direct systemic risk linkages [45]. Among the SSE's incremental risk to other markets, KOSPI (-0.549) has the highest risk, indicating that China market volatility has the strongest incremental risk to the Korean market; JKSE (-0.531) and TWII (-0.523) are also significantly affected, while KLCI (-0.285) shows the lowest sensitivity to China market volatility. In terms of relative contribution, SET (32.174%) has the highest relative risk contribution to SSE, followed by TWII (27.421%) and JKSE (25.307%), while BSE (18.624%) has the lowest; SSE has the highest risk contribution to JKSE (30.803%) and TWII (27.122%), while it has the lowest risk contribution to PSI (19.144) had the lowest risk contribution. This indicates that the Thai market plays a crucial role in transmitting risk to Chinese market, which in turn has the strongest systemic risk propagation effect on Korea (KOSPI) and Indonesia (JKSE).

To further analyze the directionality and asymmetry of systemic risk transmission, this paper employs two key metrics: the absolute difference in systemic risk contribution (ΔCoVaR^SSE - ΔCoVaR^i) and the relative difference in systemic risk contribution (%CoVaR^SSE - %CoVaR^i). The corresponding results are presented in Fig 1.

The absolute difference metric (ΔCoVaR^SSE - ΔCoVaR^i) quantifies the net impact of systemic risk transmission between SSE and other markets, making it particularly useful for assessing how systemic risk propagates from individual markets to SSE. Negative values (red arrows) indicate that a market contributes more systemic risk to SSE, classifying it as a 'Risk Contributor,' meaning SSE is more affected by it. Examples include KLCI (-0.209), SET (-0.199), PSI (-0.069), and TWII (-0.035). Conversely, positive values (blue arrows) indicate that SSE contributes more to the systemic risk of a given market, classifying it as a 'Risk Receiver,' meaning the market is more susceptible to SSE's risk transmission. Examples include KOSPI (0.143), BSE (0.099), and JKSE (0.016).

In contrast, the relative difference metric (%CoVaR^SSE - %CoVaR^i) normalizes $\Delta CoVaR$ to capture the relative strength of systemic risk propagation, making it particularly suitable for cross-market comparisons. Negative values (red arrows) indicate that a market contributes more systemic risk to SSE in relative terms, classifying it as a 'Relative Risk Contributor.' For instance, KOSPI (-4.239%), BSE (-6.049%), JKSE (-5.496%), and KLCI (-1.860%) exert stronger

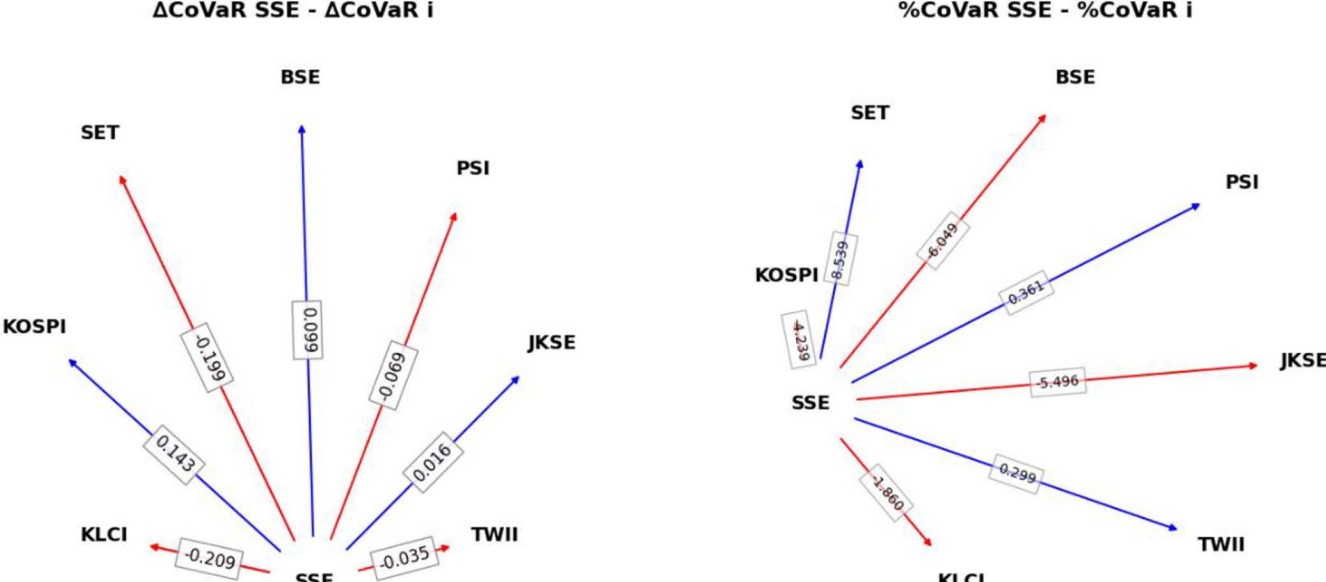

**Fig 1. Network Analysis of Systemic Risk Contributions: Absolute vs. Relative Perspectives.**

systemic risk shocks on SSE during systemic events. Positive values (blue arrows) indicate that SSE contributes more systemic risk to a market in relative terms, classifying it as a 'Relative Risk Receiver.' This is observed in SET (8.539%), PSI (0.361%), and TWII (0.299%), suggesting that SSE's market volatility has a more pronounced impact on these markets.

It is crucial to note that while $\Delta CoVaR$ quantifies the absolute magnitude of systemic risk contribution, %CoVaR captures the asymmetry in risk transmission pathways. By integrating these two indicators, we can gain a more comprehensive understanding of both the directional flow of systemic risk and the asymmetric nature of its propagation across markets, particularly in the aftermath of major uncertainties.

Fig 2 illustrates that during the 2008 global financial crisis, $\Delta CoVaR$ values exhibited a sharp decline immediately after the Lehman Brothers bankruptcy, underscoring the impact of capital flow volatility on systemic risk [46]. Specifically, Thailand (SETI) and Taiwan (TWII) demonstrated high sensitivity to risk originating from the Chinese market, which can be attributed to their trade dependence on China and vulnerability to external shocks [47]. Furthermore, the pronounced risk transmission from China's SSE to South Korea (KOSPI) and Indonesia (JKSE) suggests that China progressively emerged as a regional risk hub during the crisis [48].

$\Delta CoVaR$ values during the COVID-19 outbreak indicate that supply chain disruptions and trade dependencies were primary drivers of systemic risk propagation. In the early stages of the pandemic (January to March 2020), significant risk spillovers flowed from Malaysia (KLCI) and Indonesia (JKSE) into the Chinese market. In contrast, later in the pandemic, risk spillovers from China's SSE intensified toward export-oriented economies such as South Korea (KOSPI) and Taiwan (TWII) [49].

$\Delta CoVaR$ results from the Russia-Ukraine war indicate that, while the war's immediate impact was concentrated in energy and commodities markets, its financial repercussions extended to Asian economies via trade uncertainty and supply chain disruptions [50]. Notably, the $\Delta CoVaR$ values for Taiwan (TWII) and South Korea (KOSPI) increased markedly during the initial years of the conflict, underscoring these markets' heightened susceptibility to energy price volatility and geopolitical uncertainty. Additionally, variations in the $\Delta CoVaR$ time series reveal significant asymmetries in systemic risk transmission. For instance, risk spillovers from China's SSE to Taiwan and South Korea were notably

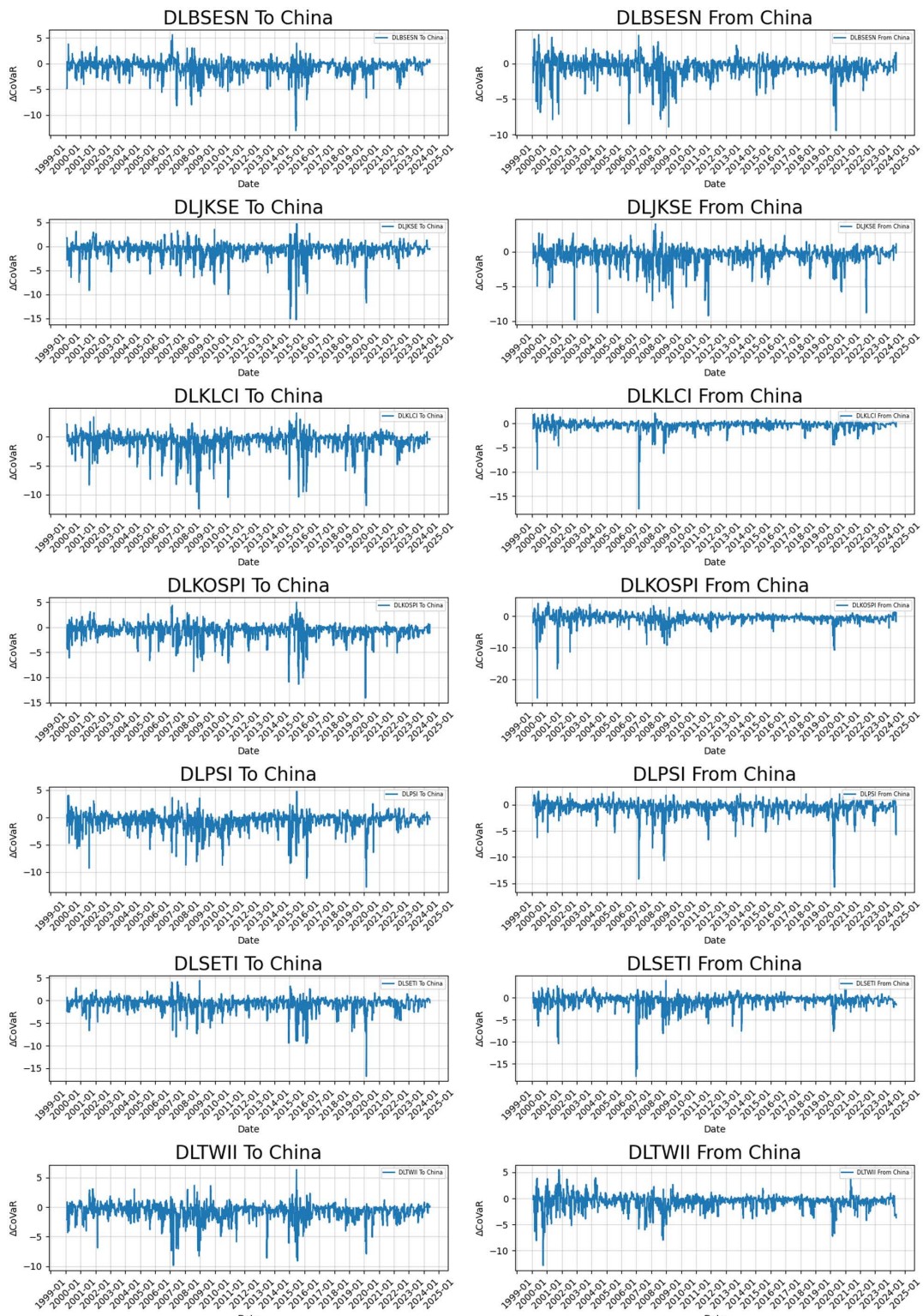

**Fig 2. Time-Varying ΔCoVaR between Emerging Markets and China's Shanghai Stock Exchange (SSE).**

stronger than the reverse, a pattern primarily driven by regional differences in trade structures and capital flows. Conversely, Malaysia and Thailand exhibited higher $\Delta CoVaR$ values for China, indicating a stronger reliance on the Chinese market during crises.

From a historical standpoint, during the 2000 dot-com bubble collapse, China was not yet fully integrated into the global financial system, resulting in limited risk spillovers to South Korea (KOSPI) and Taiwan (TWII). However, Indonesia (JKSE) and Malaysia (KLCI) had greater risk inputs into the Chinese market due to shifts in global trade dynamics [51]. Following the 2015 Chinese stock market crisis, China's SSE played an increasingly dominant role in systemic risk transmission, with spillovers to Taiwan and South Korea expanding significantly. $\Delta CoVaR$ results further suggest that China transitioned from a 'risk taker' to a 'risk transmitter' as its financial market integration deepened [52]. This period also exhibited a clear asymmetry in systemic risk spillovers, with China exporting significantly more risk to neighboring markets than it absorbed in return.

Fig 3 showed that %CoVaR rises steadily across all regions as the quantile decreases, indicating that systemic risk spillovers intensify under more extreme conditions. At q = 0.05, the conditional risk spillovers from the China's Shanghai Stock Exchange (SSE) are relatively evenly distributed among regions. Thailand (SETI) emerges as the main contributor to systemic risk for SSE, while Taiwan's TWII, Indonesia's JKSE, and Malaysia's KLSE exhibit moderate contributions. In contrast, India (BESN), Korea (KOSPI), and the Philippines (PSI) show slightly lower sensitivities, with contributions below 20%. However, a clear divergence becomes apparent at q = 0.01 (extreme risk scenarios). Notably, Taiwan (TWII) displays the highest %CoVaR, followed by Thailand (SETI)> Indonesia (JKSE)> Malaysia (KLSE)> India (BESN)> Korea (KOSPI)> the Philippines (PSI). Interestingly, some countries, such as Korea (KOSPI), exhibit decreasing and increasing contributions (albeit with minor fluctuations) from q = 0.05 to q = 0.03. This behaviour suggests that the risk transmission from KOSPI to SSE involves clear non-linear changes and a "threshold effect." When tail risks intensify and systemic pressure increases, the market might use hedging or buffering mechanisms. However, under more extreme risk conditions, the spillover effects re-escalate.

Fig 4 represents the reverse spillover effect of Fig 3, where the China's Shanghai Stock Exchange (SSE) acts as a systemic risk contributor to other Asian emerging markets. Under extreme market stress conditions at q = 0.01, all regions exhibit a higher dependency on SSE, highlighting the increased vulnerability of these financial systems in tail-risk scenarios. For instance, Indonesia (JKSE) experiences a significant impact from SSE's risk spillover under extreme conditions, making it the most affected market. This is followed by India (BESN)> Taiwan (TWII)> Thailand (SETI)> Malaysia (KLSE)> the Philippines (PSI)> Korea (KOSPI). These rankings emphasize the varying degrees of reliance and sensitivity of these markets to SSE's systemic risk spillovers during heightened market pressure.

## 5. Conclusion

This study provides a comprehensive examination of systemic risk spillovers between the China's Shanghai Stock Exchange (SSE) and seven major Asian emerging markets, employing the CoVaR and quantile regression methodologies. Our findings demonstrate the asymmetric nature of risk transmission, with certain markets exerting stronger influences on SSE and others being more vulnerable to its systemic risks. Notably, the Korean (KOSPI) market shows heightened sensitivity to SSE risk spillovers, whereas Malaysia (KLCI) exerts the most significant impact on the SSE's systemic risk. Additionally, the Thai (SET) and Taiwanese (TWII) markets emerge as substantial risk contributors and receivers, underscoring their interconnectedness with the Chinese market. These results underscore the intricate and regionally variable dynamics of systemic risk in Asia, shedding light on the importance of understanding market interdependencies in an increasingly integrated global financial environment [35,53].

Policy recommendations based on strengthening systemic risk management and market resilience building can be divided into the following two parts:

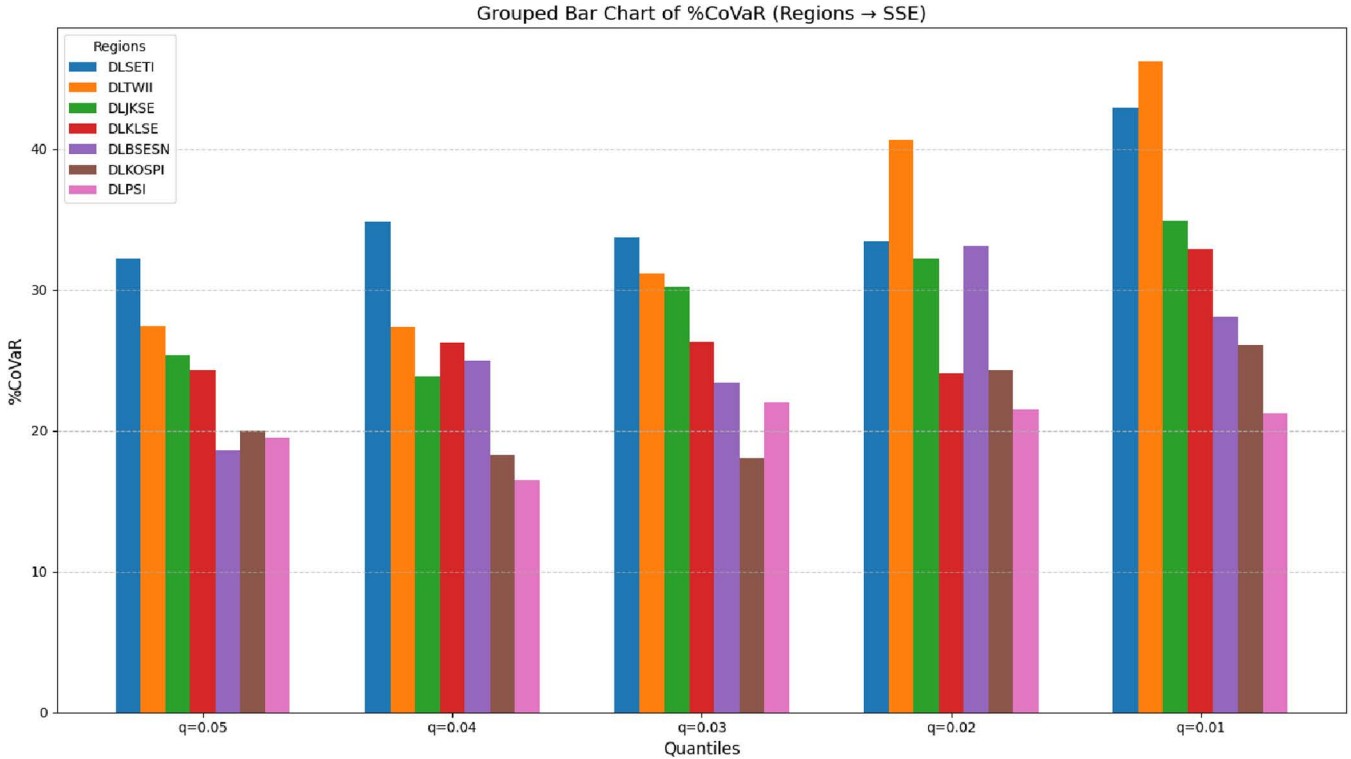

**Fig 3. %CoVaR of Asian Emerging Markets to China's Shanghai Stock Exchange (SSE).**

1) Enhancing systemic risk monitoring and regional financial stability mechanisms. This study finds that Korea (KOSPI) and Taiwan (TWII) are the most sensitive to market risk in China (SSE), while Malaysia (KLCI) and Thailand (SET) have the greatest impact on systemic risk in SSE. This phenomenon suggests that there are complex spillover effects of systemic risk in Asian emerging markets, and a more comprehensive regional financial risk monitoring and stabilisation mechanism needs to be established to enhance overall financial resilience [54]. For example, it is recommended to establish an 'Asian financial systemic risk monitoring network', integrate the data resources of financial regulators of various countries, and build a cross-border financial risk early-warning system using big data and artificial intelligence technology, so as to track market fluctuations in real time and identify the path of systemic risk contagion. In addition, financial data sharing and cross-border regulatory cooperation among Asian countries should be promoted, especially in the areas of capital flows, institutional leverage levels, and financial derivatives trading, so as to enhance the ability to identify market risks and avoid the accumulation of systemic risks.

At the same time, it is recommended to make use of the multilateral cooperation frameworks of "BRICS and the Belt and Road Initiative (BRI)" to deepen regional financial market collaboration, enhance financial infrastructure and reduce market volatility. Financial cooperation under the BRICS New Development Bank (NDB) and the Belt and Road Initiative (BRI) can serve as an important pillar of regional financial stability, promote financial regulatory coordination among relevant countries/regions, and improve the defence capability against global capital market turbulence.

2) Optimising capital buffers, controlling cross-border capital flows and promoting market diversification

The high degree of interconnectedness of Asian emerging markets dictates that financial risks are not confined to a single market, but are transmitted through multiple channels such as capital flows, investor behaviour, exchange rate

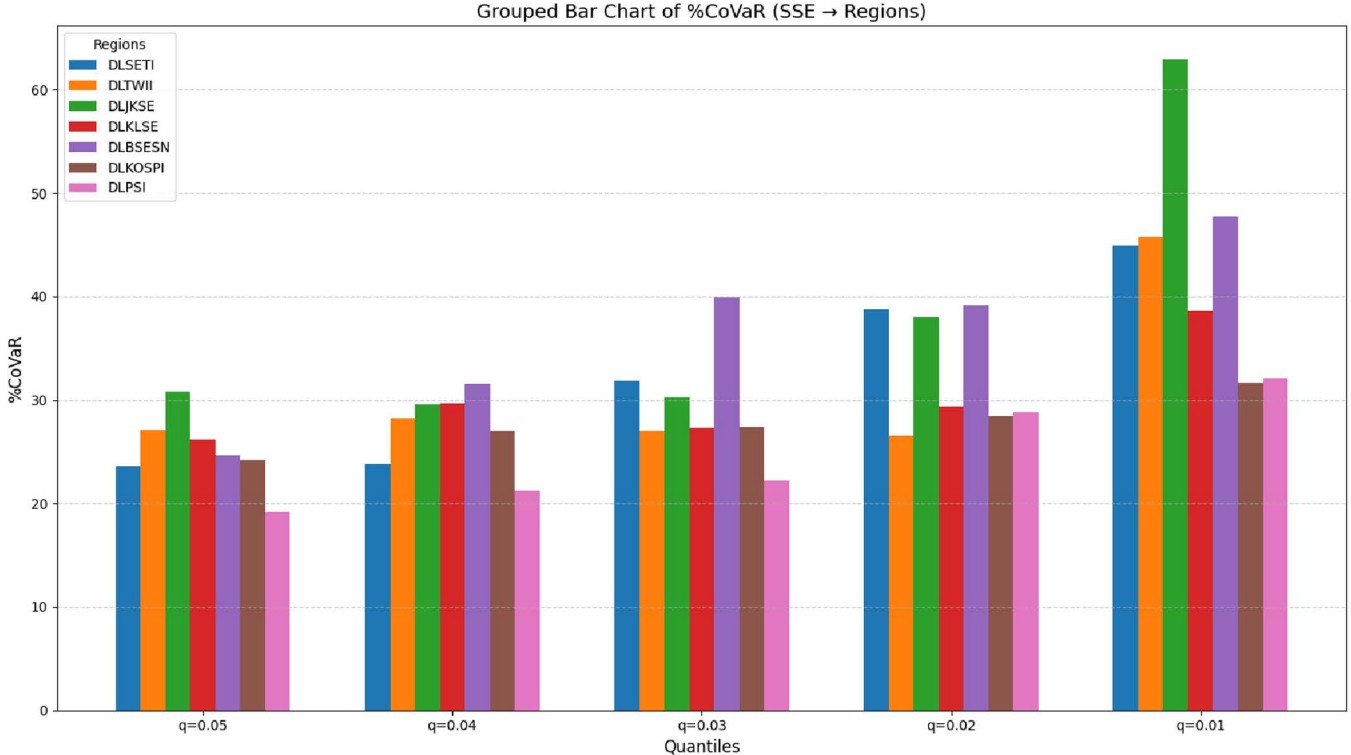

**Fig 4. %CoVaR of China's Shanghai Stock Exchange (SSE) to Asian Emerging Markets.**

volatility, etc. [55]. Therefore, the macroprudential management framework should be strengthened to optimise the capital buffer mechanism, while promoting capital market diversification to reduce the risk of market homogeneity.

In terms of capital management, South Korea (KOSPI) and Taiwan (TWII), as the most sensitive markets to SSE risk, need to strengthen the capital buffer mechanism in the banking and securities markets, and increase the capital adequacy requirement to enhance the ability to cope with SSE market volatility. In addition, for markets with strong risk transmission, such as Malaysia (KLCI) and Thailand (SET), it is recommended to introduce macro-prudential capital flow management tools (e.g., short-term capital flow tax, minimum holding period for capital, etc.) to avoid short-term capital outflows from exacerbating market turbulence in times of severe market volatility. Meanwhile, in terms of market diversification construction, countries should promote the synergistic development of bond markets and stock markets to reduce the systemic shock caused by a single market crash. In addition, regulators can promote institutional investors (e.g., pension funds, insurance companies) to expand their positions in asset classes such as bonds, alternative investments, and commodities to reduce the impact of stock market volatility on the overall financial market.

Finally, it is recommended to use fintech to enhance market resilience by promoting AI-driven market risk monitoring systems combined with smart melting mechanisms to improve market stability [56]. Under extreme market stress, intelligent trading restrictions can be implemented to prevent liquidity crises from occurring through automated liquidity support tools that inject funds when market liquidity is tight.

In summary, emerging markets in Asia should reduce the contagion effect of cross-border financial risks, improve overall market resilience and promote Strengthened regional financial cooperation, such as information sharing and coordinated risk management strategies, which can also play a pivotal role in mitigating systemic risks [9]. Despite its contributions, this study is not without limitations. First, the analysis is constrained by its reliance on historical data, which may not

fully capture evolving market dynamics and structural changes in financial systems. Second, the study focuses exclusively on equity markets, omitting other financial sectors such as bond or foreign exchange markets, which may exhibit different risk transmission mechanisms. Third, while the CoVaR methodology effectively quantifies risk spillovers, it does not account for nonlinear dependencies or higher-order systemic interactions among multiple markets. These limitations point to potential areas for methodological enhancement and broader scope in future research. For instance, integrating cross-sectoral data and exploring the role of financial institutions could provide a more comprehensive understanding of systemic risks. Moreover, advanced modeling techniques such as machine learning could complement traditional econometric approaches to better capture dynamic and nonlinear relationships.

## Author contributions

**Conceptualization:** Zhongzheng Fang.

**Data curation:** Zhongzheng Fang.

**Formal analysis:** Zhongzheng Fang.

**Methodology:** Zhongzheng Fang.

**Software:** Zhongzheng Fang.

**Validation:** Zhongzheng Fang.

**Visualization:** Zhongzheng Fang.

**Writing – original draft:** Zhongzheng Fang.

**Writing – review & editing:** Zhongzheng Fang.

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
