## [Decision Letter · Decision Letter 0]

4 Feb 2025

PONE-D-24-60406A Study of Systemic Risk Spillovers in Asian Emerging Markets and Chinese Stock MarketPLOS ONE

Dear Dr. Fang,

Thank you for submitting your manuscript to PLOS ONE.  In view of the referees’ feedback and my own reading of your paper, we invite you to address all issues noted below. Although the reviewers consider that the manuscript requires a minor revision, we believe that these issues are major in nature, requiring more than a superficial or minor revision. We have particular concerns about the robustness of the methods and analysis and the soundness and basis of the conclusions. On the other hand, we also consider that the literature review section is not deep enough. 

Since our point of view the paper has an important potential to be consider for publication on this journal, so we invite you to address the issues noted below and resubmit the manuscript for a new revision round. 

We look forward to receiving your revised manuscript.

Kind regards,

Juan E. Trinidad-Segovia, PhD

Section Editor

PLOS ONE

Journal requirements:   When submitting your revision, we need you to address these additional requirements. 1. Please ensure that your manuscript meets PLOS ONE's style requirements, including those for file naming. The PLOS ONE style templates can be found at https://journals.plos.org/plosone/s/file?id=wjVg/PLOSOne_formatting_sample_main_body.pdf and https://journals.plos.org/plosone/s/file?id=ba62/PLOSOne_formatting_sample_title_authors_affiliations.pdf.

Reviewers' comments:

Reviewer's Responses to Questions

**Comments to the Author**

1. Is the manuscript technically sound, and do the data support the conclusions?

Reviewer #1: Yes

Reviewer #2: Partly

2. Has the statistical analysis been performed appropriately and rigorously? 

Reviewer #1: Yes

Reviewer #2: Yes

3. Have the authors made all data underlying the findings in their manuscript fully available?

Reviewer #1: Yes

Reviewer #2: Yes

4. Is the manuscript presented in an intelligible fashion and written in standard English?

Reviewer #1: Yes

Reviewer #2: No

5. Review Comments to the Author

Reviewer #1: This manuscript explores the systemic risk spillovers between China's Shanghai Stock Exchange and seven Asian emerging markets, which is a relevant and timely topic in the context of global financial integration. The study uses appropriate methodologies and provides some valuable insights. However, there are several areas that need improvement before it can be considered for publication.

1. The literature review, while covering some relevant studies, could be more in-depth. It mainly focuses on introducing the concepts of risk spillover and the methodologies used in previous research. However, it fails to provide a more critical analysis of how these previous studies (see, e.g., https://doi.org/10.1016/j.ememar.2023.101020;
https://doi.org/10.1016/j.irfa.2023.102518) have shaped the current research question and what new contributions this study aims to make in more detail. There is also a lack of discussion on the potential differences in risk spillover mechanisms across different economic and financial environments.

2. The author studies the risk spillover effects between pairwise markets using CoVaR. However, I would like to suggest the author construct a CoVaR-based network from a systemic perspective to comprehensively explore the risk spillover effects among China’s stock markets and seven Asian emerging markets.

3. The author explores the risk spillover effects between China and seven Asian emerging markets. However, the driving mechanisms or influencing factors of these risk spillover effects are not deeply investigated. It is recommended that the author conduct additional empirical research or analysis in this regard.

Reviewer #2: Abstract Clarity: The abstract contains minor phrasing issues. For example, the sentence “the study fills a gap in understanding the asymmetric across these markets” is unclear and should be revised for better clarity and readability.

Dataset selection: The rationale for market selection aligns with MSCI classifications, but the exclusion of certain regional markets should be more clearly justified.

Risk Spillover Dynamics: While the study analyzes risk spillovers broadly, it would benefit from examining how risks propagate during specific financial crises. Although global events such as COVID-19 and the Russia-Ukraine war are mentioned, their specific impacts on systemic risk dynamics require more detailed exploration.

Literature and Emerging Markets: The literature review could be expanded to include recent studies on systemic risk and financial contagion, particularly in emerging markets.

Visual Aids: Incorporating visual aids like influence networks to illustrate risk spillovers would improve the clarity and accessibility of the findings.

Policy Recommendations: While the study suggests strengthening macroprudential frameworks, the recommendations are generic. The findings should be used to propose specific policies to mitigate systemic vulnerabilities in the region.

Typographical Errors: The paper includes typographical errors, such as inconsistencies in notations. For instance, the price at day t is referred to as Rt in the text but is denoted as P_t in the equations.

6. PLOS authors have the option to publish the peer review history of their article (what does this mean? ). If published, this will include your full peer review and any attached files.

**Do you want your identity to be public for this peer review?** For information about this choice, including consent withdrawal, please see our Privacy Policy .

Reviewer #1: No

Reviewer #2: No

---

## [Author Response · Author response to Decision Letter 1]

25 Feb 2025

We sincerely appreciate the time and effort of the editor and reviewers in evaluating our manuscript. We are grateful for the constructive feedback, which has significantly helped us improve the quality and clarity of our research. We have carefully addressed each of the concerns raised and incorporated necessary revisions throughout the manuscript.

Below is a point-by-point response to the editor’s and reviewers’ comments:

Response to reviewer comment #1:

1. Literature review needs more depth

The revised article has expanded the literature review by adding critical analysis, integrating recent research, and more clearly articulating the contributions of this study.

2. Building a CoVaR-based systems analysis network

Added Figure 1, which illustrates a CoVaR-based systematic risk contribution network analysis, distinguishing between risk contributors and risk recipients.

3. Investigating the driving mechanisms of risk spillovers

Added Figure 2, which provides a time-varying ΔCoVaR analysis to assess risk spillovers during major crises and explain the economic mechanisms behind them.

Response to reviewer comment #2:

1. Clarity of the abstract

Revised the abstract for clarity, in particular correcting the unclear phrase “the study fills a gap in the understanding of these market asymmetries”.

2. Rationale for dataset selection

Market selection based on MSCI classification, financial openness and market liquidity is explained in detail.

3. Risk spillover dynamics during crises

Integrates Figure 2 to analyze spillover patterns during major crises, explaining their economic drivers and financial implications.

4. Expanding the literature on risk spillovers from emerging markets

Adds recent literature on systemic risk transmission, with a particular focus on financial contagion in emerging markets.

5. Inclusion of visual aids to risk transmission

Added Figure 1, which illustrates the systemic risk network using CoVaR-based spillover indicators.

6. Specific policy recommendations

Revised the policy section with specific recommendations, including

1) Establish an “Asian Financial Systemic Risk Monitoring Network”.

2) Implement macro-prudential capital flow management tools.

3) Utilize artificial intelligence-driven market risk monitoring systems to promote financial stability.

7. Typographical and notation errors

Corrected all typographical and notational inconsistencies.

---

## [Decision Letter · Decision Letter 1]

21 Mar 2025

A Study of Systemic Risk Spillovers in Asian Emerging Markets and Chinese Stock Market

PONE-D-24-60406R1

Dear Dr. Fang,

We’re pleased to inform you that your manuscript has been judged scientifically suitable for publication and will be formally accepted for publication once it meets all outstanding technical requirements.

Kind regards,

Juan E. Trinidad-Segovia, PhD

Section Editor

PLOS ONE

Additional Editor Comments (optional):

Reviewers' comments:

Reviewer's Responses to Questions

**Comments to the Author**

1. If the authors have adequately addressed your comments raised in a previous round of review and you feel that this manuscript is now acceptable for publication, you may indicate that here to bypass the “Comments to the Author” section, enter your conflict of interest statement in the “Confidential to Editor” section, and submit your "Accept" recommendation.

Reviewer #2: All comments have been addressed

2. Is the manuscript technically sound, and do the data support the conclusions?

Reviewer #2: Yes

3. Has the statistical analysis been performed appropriately and rigorously? 

Reviewer #2: Yes

4. Have the authors made all data underlying the findings in their manuscript fully available?

Reviewer #2: Yes

5. Is the manuscript presented in an intelligible fashion and written in standard English?

Reviewer #2: Yes

6. Review Comments to the Author

Reviewer #2: (No Response)

7. PLOS authors have the option to publish the peer review history of their article (what does this mean? ). If published, this will include your full peer review and any attached files.

**Do you want your identity to be public for this peer review?** For information about this choice, including consent withdrawal, please see our Privacy Policy .

Reviewer #2: No

---

## [Editor Report · Acceptance letter]

PONE-D-24-60406R1

PLOS ONE

Dear Dr. Fang,

I'm pleased to inform you that your manuscript has been deemed suitable for publication in PLOS ONE. Congratulations! Your manuscript is now being handed over to our production team.

Kind regards,

on behalf of

Dr. Juan E. Trinidad-Segovia

Section Editor

PLOS ONE